# Efficient Discretization of Optimal Transport

**DOI:** 10.3390/e25060839

**Published:** 2023-05-24

**Authors:** Junqi Wang, Pei Wang, Patrick Shafto

**Affiliations:** 1Department of Math & CS, Rutgers University, Newark, NJ 07102, USA; junqi.wang@rutgers.edu (J.W.); peiwang@rutgers.edu (P.W.); 2School of Mathematics, Institute for Advanced Study, Princeton, NJ 08540, USA

**Keywords:** optimal transport, entropy regularization, discretization, gradient descent

## Abstract

Obtaining solutions to optimal transportation (OT) problems is typically intractable when marginal spaces are continuous. Recent research has focused on approximating continuous solutions with discretization methods based on i.i.d. sampling, and this has shown convergence as the sample size increases. However, obtaining OT solutions with large sample sizes requires intensive computation effort, which can be prohibitive in practice. In this paper, we propose an algorithm for calculating discretizations with a given number of weighted points for marginal distributions by minimizing the (entropy-regularized) Wasserstein distance and providing bounds on the performance. The results suggest that our plans are comparable to those obtained with much larger numbers of i.i.d. samples and are more efficient than existing alternatives. Moreover, we propose a local, parallelizable version of such discretizations for applications, which we demonstrate by approximating adorable images.

## 1. Introduction

Optimal transport is the problem of finding a coupling of probability distributions that minimizes cost [1], and it is a technique applied across various fields and literatures [2,3]. Although many methods exist for obtaining optimal transference plans for distributions on discrete spaces, computing the plans is not generally possible for continuous spaces [4]. Given the prevalence of continuous spaces in machine learning, this is a significant limitation for theoretical and practical applications.

One strategy for approximating continuous OT plans is based on discrete approximation via sample points. Recent research has provided guarantees on the fidelity of discrete, sample-location-based approximations for continuous OT as the sample size N→∞ [5]. Specifically, by sampling large numbers of points $S_{i}$ from each marginal, one may compute a discrete optimal transference plan on $S_{1}\times S_{2}$, with the cost matrix being derived from the pointwise evaluation of the cost function on $S_{1}\times S_{2}$.

Even in the discrete case, obtaining minimal cost plans is computationally challenging. For example, Sinkhorn scaling, which computes an entropy-regularized approximation for OT plans, has a complexity that scales with $|S_{1}\times S_{2}|$ [6]. Although many comparable methods exist [7], all of them have a complexity that scales with the product of sample sizes, and they require the construction of a cost matrix that also scales with $|S_{1}\times S_{2}|$.

We have developed methods for optimizing both sampling locations and weights for small N approximations of OT plans (see Figure 1). In Section 2, we formulate the problem of fixed size approximation and reduce it to discretization problems on marginals with theoretical guarantees. In Section 3, the gradient of entropy-regularized Wasserstein distance between a continuous distribution and its discretization is derived. In Section 4, we present a stochastic gradient descent algorithm that is based on the optimization of the locations and weights of the points with empirical demonstrations. Section 5 introduces a parellelizable algorithm via decompositions of the marginal spaces, which reduce the computational complexity by exploiting intrinsic geometry. In Section 6, we analyze time and space complexity. In Section 7, we illustrate the advantage of including weights for sample points by providing a comparison with an existing location that is only based on discretization.

## 2. Efficient Discretizations

Optimal transport (OT): Let $\left(X,d_{X}\right)$, $\left(Y,d_{Y}\right)$ be compact Polish spaces (complete separable metric spaces), μ∈P(X), ν∈P(Y) be probability distributions on their Borel-algebras, and c:X×Y→R be a cost function. Denote the set of all joint probability measures (couplings) on X×Y with marginals μ and ν by Π(μ,ν). For the cost function *c*, the optimal transference plan between μ and ν is defined as in [1]: $\gamma \left(\mu ,\nu \right)\mathrm{argmin}{\;}_{\pi \in \Pi (\mu ,\nu )}\langle c,\pi \rangle$, where $\langle c,\pi \rangle \int_{X\times Y}c\left(x,y\right)d\pi \left(x,y\right)$.

When X=Y, the cost $c\left(x,y\right)=d_{X}^{k}\left(x,y\right)$, $W_{k}\left(\mu ,\nu \right)={\langle c,\gamma \left(\mu ,\nu \right)\rangle }^{1/k}$ defines the *k*-Wasserstein distance between μ and ν for k≥1. Here, $d_{X}^{k}\left(x,y\right)$ is the *k*-th power of the metric $d_{X}$ on *X*.

Entropy regularized optimal transport (EOT)  [5,8] was introduced to estimate OT couplings with reduced computational complexity: $\gamma_{\lambda }\left(\mu ,\nu \right):=\mathrm{argmin}{\;}_{\pi \in \Pi (\mu ,\nu )}\langle c,\pi \rangle +\lambda \mathrm{KL}\left(\pi ||\mu ⊗\nu \right)$, where λ>0 is a regularization parameter, and the regularization term KL(π||μ⊗ν):=$\int \log(\frac{d\pi }{d\mu ⊗d\nu })d\pi$ is the Kullback–Leibler divergence. The EOT objective is smooth and convex, and its unique solution with a given discrete (μ,ν,c) can be obtained using a Sinkhorn iteration (SK)  [9].

However, for large-scale discrete spaces, the computational cost of SK can still be unfeasible [6]. Even worse, to even apply the Sinkhorn iteration, one must know the entire cost matrix over the large-scale spaces, which itself can be a non-trivial computational burden to obtain; in some cases, for example, where the cost is derived from a probability model [10], it may require intractable computations [11,12].

The Framework: We propose the optimization of the location and weights of a fixed size discretization to estimate the continuous OT. The discretization on X×Y is completely determined by those on *X* and *Y* to respect the marginal structure in the OT. Let $m,n\in Z^{*}$, $\mu_{m}\in P\left(X\right)$, $\nu_{n}\in P\left(Y\right)$ be a discrete approximation of μ and ν, respectively, with $\mu_{m}=\sum_{i=1}^{m}w_{i}\delta_{x_{i}}$, $\nu_{n}=\sum_{j=1}^{n}u_{j}\delta_{y_{j}}$, $x_{i}\in X$, $y_{j}\in Y$, and $w_{i},u_{j}\in R^{+}$. Then, the EOT plan $\gamma_{\lambda }\left(\mu ,\nu \right)\in \Pi \left(\mu ,\nu \right)$ for the OT problem (μ,ν,c) can be approximated by the EOT plan $\gamma_{\lambda }\left(\mu_{m},\nu_{n}\right)\in \Pi \left(\mu_{m},\nu_{n}\right)$ for the OT problem $\left(\mu_{m},\nu_{n},c\right)$. There are three distributions that have their discrete counterparts; thus, with a fixed size $m,n\in Z^{*}$, a naive idea about the objective to be optimized can be
(1)$$\begin{array}{c} \Omega_{k,\rho }\left(\mu_{m},\nu_{n}\right)=W_{k}^{k}\left(\mu ,\mu_{m}\right)+W_{k}^{k}\left(\nu ,\nu_{n}\right)+\rho W_{k}^{k}\left(\gamma_{\lambda }\left(\mu ,\nu \right),\gamma_{\lambda }\left(\mu_{m},\nu_{n}\right)\right), \end{array}$$
where $W_{k}^{k}\left(\phi ,\psi \right)$ represents the *k*-th power of *k*-Wasserstein distance between measures ϕ and ψ. The hyperparameter ρ>0 balances between the estimation accuracy over marginals and that of the transference plan, while the weights on marginals are equal.

To properly compute $W_{k}^{k}\left(\gamma_{\lambda }\left(\mu ,\nu \right),\gamma_{\lambda }\left(\mu_{m},\nu_{n}\right)\right)$, a metric $d_{X\times Y}$ on X×Y is needed. We expect $d_{X\times Y}$ on *X*-slices or *Y*-slices to be compatible with $d_{X}$ or $d_{Y}$, respectively; furthermore, we may assume that there exists a constant A>0 such that:(2)$$\begin{array}{c} \max\left\{d_{X}^{k}\left(x_{1},x_{2}\right),d_{Y}^{k}\left(y_{1},y_{2}\right)\right\}\leq d_{X\times Y}^{k}\left(\left(x_{1},y_{1}\right),\left(x_{2},y_{2}\right)\right)\leq A\left(d_{X}^{k}\left(x_{1},x_{2}\right)+d_{Y}^{k}\left(y_{1},y_{2}\right)\right). \end{array}$$For instance, (Equation 2) holds when $d_{X\times Y}$ is the *p*-product metric for 1≤p≤∞.

The objective $\Omega_{k,\rho }\left(\mu_{m},\nu_{n}\right)$ is estimated by its entropy regularized approximation $\Omega_{k,\zeta ,\rho }\left(\mu_{m},\nu_{n}\right)$ for efficient computation, where ζ is the regularization parameter, as follows:(3)$$\begin{array}{c} \Omega_{k,\zeta ,\rho }\left(\mu_{m},\nu_{n}\right)=W_{k,\zeta }^{k}\left(\mu ,\mu_{m}\right)+W_{k,\zeta }^{k}\left(\nu ,\nu_{n}\right)+\rho W_{k,\zeta }^{k}\left(\gamma_{\lambda }\left(\mu ,\nu \right),\gamma_{\lambda }\left(\mu_{m},\nu_{n}\right)\right). \end{array}$$Here, $W_{k}^{k}\left(\mu ,\mu_{m}\right)={\langle d_{X}^{k},\gamma \left(\mu ,\mu_{m}\right)\rangle }^{1/k}$ is estimated by $W_{k,\zeta }^{k}\left(\mu ,\mu_{m}\right)={\langle d_{X}^{k},\gamma_{\zeta }\left(\mu ,\mu_{m}\right)\rangle }^{1/k}$. $\gamma_{\zeta }\left(\mu ,\mu_{m}\right)$ is computed by optimizing ${\hat{W}}_{k,\zeta }^{k}\left(\mu ,\mu_{m}\right)\;=\;\langle d_{X}^{k},\gamma_{\zeta }\left(\mu ,\mu_{m}\right)\rangle +\lambda \mathrm{KL}(\gamma_{\zeta }\left(\mu ,\mu_{m}\right)||\mu ⊗\mu_{m}).$

One major difficulty in optimizing $\Omega_{k,\zeta ,\rho }\left(\mu_{m},\nu_{n}\right)$ is to evaluate $W_{k,\zeta }^{k}\left(\gamma_{\lambda }\left(\mu ,\nu \right),\gamma_{\lambda }\left(\mu_{m},\nu_{n}\right)\right)$. In fact, obtaining $\gamma_{\lambda }\left(\mu ,\nu \right)$ is intractable, which is the original motivation for the discretization. To overcome this drawback, by utilizing the dual formulation of EOT, the following are shown (see proof in Appendix A):

**Proposition** **1.**
*When X and Y are two compact spaces, and the cost function c is $C^{\infty }$, there exists a constant $C_{1}\in R^{+}$ such that*

$$\begin{array}{c} \max\left\{W_{k}^{k}\left(\mu ,\mu_{m}\right),\;W_{k}^{k}\left(\nu ,\nu_{n}\right)\right\}\;\leq \;W_{k,\zeta }^{k}\;\left(\gamma_{\lambda }\left(\mu ,\nu \right),\;\gamma_{\lambda }\left(\mu_{m},\nu_{n}\right)\right)\leq C_{1}\left[W_{k,\zeta }^{k}\left(\mu ,\mu_{m}\right)+W_{k,\zeta }^{k}\left(\nu ,\nu_{n}\right)\right]. \end{array}$$



Notice that Proposition 1 indicates that $W_{k,\zeta }^{k}\;\left(\gamma_{\lambda }\left(\mu ,\nu \right),\;\gamma_{\lambda }\left(\mu_{m},\nu_{n}\right)\right)$ is bounded above by multiples of $W_{k,\zeta }^{k}\left(\mu ,\mu_{m}\right)+W_{k,\zeta }^{k}\left(\nu ,\nu_{n}\right)$, i.e., when the continuous marginals μ and ν are properly approximated, so is the optimal transference plan between them. Therefore, to optimize $\Omega_{k,\zeta ,\rho }\left(\mu_{m},\nu_{n}\right)$, we focus on developing algorithms to obtain $\mu_{m}^{*},\nu_{n}^{*}$ that minimize $W_{k,\zeta }^{k}\left(\mu ,\mu_{m}\right)$ and $W_{k,\zeta }^{k}\left(\nu ,\nu_{n}\right)$.

**Remark** **1.**
*The regularizing parameters (λ and ζ above) introduce smoothness, together with an error term, into the OT problem. To make an accurate approximation, we need λ and ζ to be as small as possible. However, when parameters become too small, the matrices to be normalized in the Sinkhorn algorithm lead to an overflow or underflow problem of numerical data types (32-bit or 64-bit floating point numbers). Thus, the value for regularizing the constant threshold is proportional to the k-th power of the diameter of the supported region. In this work, we try our best to control the value (mainly on ζ), which ranges from 10^−4^ to 0.01 when the diameter is 1 in different examples.*


## 3. Gradient of the Objective Function

Let $\nu =\sum_{i=1}^{m}w_{i}\delta_{y_{i}}$ be a discrete probability measure in the position of “$\mu_{m}$” in the last section. For a fixed (continuous) μ, the objective now is to obtain a discrete target $\nu^{*}=\mathrm{argmin}\;W_{k,\zeta }^{k}\left(\mu ,\nu \right)$.

In order to apply a stochastic gradient descent (SGD) to both the positions ${\left\{y_{i}\right\}}_{i=1}^{m}$ and their weights ${\left\{w_{i}\right\}}_{i=1}^{m}$ to achieve $\nu^{*}$, we now derive the gradient of $W_{k,\zeta }^{k}\left(\mu ,\nu \right)$ about ν by following the discrete discussions of [13,14]. The SGD on *X* is either derived through an exponential map, or by treating *X* as (part of) an Euclidean space.

Let $g\left(x,y\right):=d_{X}^{k}\left(x,y\right)$, and denote the joint distribution minimizing ${\hat{W}}_{k,\zeta }^{k}$ as π with the differential form at $\left(x,y_{i}\right)$ being $d\pi_{i}\left(x\right)$, which is used to define $W_{k,\zeta }^{k}$ in Section 2.

By introducing the Lagrange multipliers $\alpha \in L^{\infty }\left(X\right),\beta \in R^{m}$i, we have ${\hat{W}}_{k,\zeta }^{k}\left(\mu ,\nu \right)=\max_{\alpha ,\beta }$L(μ,ν;α,β), where $L\left(\mu ,\nu ;\alpha ,\beta \right)=\int_{X}\alpha \left(x\right)d\mu \left(x\right)+\sum_{i=1}^{n}\beta w_{i}-\zeta \int_{X}\sum_{i=1}^{n}w_{i}E_{i}\left(x\right)d\mu \left(x\right)$ with $E_{i}\left(x\right)=e^{(\alpha \left(x\right)+\beta_{i}g\left(x,y_{i}\right))/\zeta }$ (see [5]). Let $\alpha^{*},\beta^{*}$ be the argmax; then, we have
$$W_{k,\zeta }^{k}\left(\mu .\nu \right)=\int_{X}\sum_{i=1}^{n}g\left(x,y_{i}\right)E_{i}^{*}\left(x\right)w_{i}d\mu \left(x\right)$$
with $E_{i}^{*}\left(x\right)=e^{(\alpha^{*}\left(x\right)+\beta_{i}^{*}-g\left(x,y_{i}\right))/\zeta }$. Since $\alpha^{\prime }\left(x\right):=\alpha \left(x\right)+t$ and $\beta_{i}^{\prime }:=\beta_{i}-t$ produce the same $E_{i}\left(x\right)$ for any t∈R, the representative with $\beta_{n}=0$ that is equivalent to β (as well as $\beta^{*}$) is denoted by $\bar{\beta }$ (similarly ${\bar{\beta }}^{*}$) below in order to obtain uniqueness and make the differentiation possible.

From a direct differentiation of $W_{k,\zeta }^{k}$, we have
(4)$$\begin{array}{l} \frac{\partial W_{k,\zeta }^{k}}{\partial w_{i}}=\int_{X}g\left(x,y_{i}\right)E_{i}^{*}\left(x\right)d\mu \left(x\right)+\\ \frac{1}{\zeta }\int_{X}\sum_{j=1}^{n}g\left(x,y_{j}\right)\left(\frac{\partial \alpha^{*}\;\left(x\right)}{\partial w_{i}}+\frac{\partial \beta_{j}^{*}}{\partial w_{i}}\right)w_{j}E_{j}^{*}\left(x\right)d\mu \left(x\right). \end{array}$$
(5)$$\begin{array}{c} \nabla_{y_{i}}\;W_{k,\zeta }^{k}=\;\int_{X}\;\;\;\nabla_{y_{i}}g\left(x,y_{i}\right)\left(\;1-\frac{g(x,y_{i})}{\zeta }\;\right)\;E_{i}^{*}\left(x\right)w_{i}d\mu \left(x\right)+\\ \frac{1}{\zeta }\;\;\int_{X}\sum_{j=1}^{n}g\left(x,y_{j}\right)\;\left(\nabla_{y_{i}}\;\alpha^{*}\;\left(x\right)\;+\;\nabla_{y_{i}}\;\beta_{j}^{*}\right)w_{j}E_{j}^{*}\left(x\right)d\mu \left(x\right). \end{array}$$With the transference plan $d\pi_{i}\left(x\right)=w_{i}E_{i}^{*}\left(x\right)d\mu \left(x\right)$ and the derivatives of $\alpha^{*}$, $\beta^{*}$, $g(x,y_{i})$ calculated, the gradient of $W_{k,\zeta }^{k}$ can be assembled.

Assume that *g* is a Lipschitz constant that is differentiable almost everywhere (for k≥1 and a $d_{X}$ Euclidean distance in $R^{d}$, differentiability fails to hold only when k=1 and $y_{i}=x$) and that $\nabla_{y}g\left(x,y\right)$ is calculated. The derivatives of $\alpha^{*}$ and ${\bar{\beta }}^{*}$ can then be calculated thanks to the Implicit Function Theorem for Banach spaces (see [15]).

The maximality of L at $\alpha^{*}$ and ${\bar{\beta }}^{*}$ induces $N:=\nabla_{\alpha ,\bar{\beta }}{L|}_{\left(\alpha^{*},{\bar{\beta }}^{*}\right)}=0\in {\left(L^{\infty }\left(X\right)⊗R^{m-1}\right)}^{\vee }$, and the Fréchet derivative vanishes. By differentiating (in the sense of Fréchet) again on $\left(\alpha ,\bar{\beta }\right)$ and $y_{i},w_{i}$, respectively, we get
(6)$$\nabla_{\left(\alpha ,\bar{\beta }\right)}N=-\frac{1}{\zeta }\left[\begin{array}{cc} d\mu \left(x\right)\delta \left(x,x^{\prime }\right) & d\pi_{j}\left(x^{\prime }\right)\\ d\pi_{i}\left(x\right) & w_{i}\delta_{ij} \end{array}\right]$$
as a bilinear functional on $L^{\infty }\left(X\right)\times R^{m-1}$ (note that, in Equation (Equation 6), the index *i* of $d\pi_{i}$ cannot be *m*). The bilinear functional $\nabla_{\left(\alpha ,\bar{\beta }\right)}N$ is invertible, and we denote its inverse by M as a bilinear form on ${\left(L^{\infty }\left(X\right)⊗R^{m-1}\right)}^{\vee }$. The last ingredient for the Implicit Function Theorem is $\nabla_{w_{i},y_{i}}N$:(7)$$\nabla_{w_{i}}N=\left(-\frac{1}{w_{i}}\int_{X}\left(\;\cdot \;\right)d\pi_{i}\left(x\right),\vec{0}\right)$$
(8)$$\begin{array}{c} \nabla_{y_{i}}N=\left(\frac{1}{\zeta }\int_{X}\left(\cdot \right)\nabla_{y_{i}}g\left(x,y_{i}\right)d\pi_{i}\left(x\right),\right.\left.\frac{\delta_{ij}}{\zeta }\int_{X}\nabla_{y_{i}}g\left(x,y_{i}\right)d\pi_{i}\left(x\right)\right). \end{array}$$Then, $\nabla_{w_{i},y_{i}}\left(\alpha^{*},{\bar{\beta }}^{*}\right)=M\left(\nabla_{w_{i},y_{i}}N\right)$. Therefore, we have gradient $\nabla_{w_{i},y_{i}}W_{k,\zeta }^{k}$ calculated.

Moreover, we can differentiate Equations (Equation 4)–(Equation 8) to get a Hessian matrix of $W_{k,\zeta }^{k}$ on $w_{i}$’s and $y_{i}$’s to provide a better differentiability of g(x,y) (which may enable Newton’s method, or a mixture of Newton’s method and minibatch SGD to accelerate the convergence). More details about the claims, calculations, and proofs are provided in the Appendix B.

## 4. The Discretization Algorithm

Here, we provide a description of an algorithm for the efficient discretizations of optimal transport (EDOT) from a distribution μ to $\mu_{m}$ with integer *m*, which is a given cardinality of support. In general, μ does not need not be explicitly accessible, and, even if it is accessible, computing the exact transference plan is not feasible. Therefore, in this construction, we assume that μ is given in terms of a random sampler, and we apply a minibatch stochastic gradient descent (SGD) through a set of samples that are independently drawn from μ of size *N* on each step to approximate μ.

To calculate the gradient $\nabla_{\mu_{m}}W_{k,\zeta }^{k}\left(\mu ,\mu_{m}\right)={\left(\nabla_{x_{i}}W_{k,\zeta }^{k}\left(\mu ,\mu_{m}\right),\nabla_{w_{i}}W_{k,\zeta }^{k}\left(\mu ,\mu_{m}\right)\right)}_{i=1}^{m}$, we need: (1).  πX,ζ|, the EOT transference plan between μ and $\mu_{m}$, (2).  the cost g=dX|k on *X*, and (3).  its gradient on the second variable ∇x′dX|k(x,x′). From *N* samples ${\left\{y_{i}\right\}}_{i=1}^{N}$, we can construct μN|=1N∑i=1Nδyi and calculate the gradients with μ replaced by μN| as an estimation, whose effectiveness (convergence as N→∞) is proved in [5].

We call this discretization algorithm the *Simple EDOT* algorithm. The pseudocode is stated in the Appendix C.

**Proposition** **2.**(Convergence of the Simple EDOT). *The Simple EDOT generates a sequence $\left(\mu_{m}^{\left(i\right)}\right)$ in the compact set $X^{m}\times \Delta$. If the set of limit points of $\left(\mu_{m}^{\left(i\right)}\right)$ does not intersect with $X^{m}\times \partial \Delta$, then $\left(\mu_{m}^{\left(i\right)}\right)$ converges to a stationary point in $X^{m}\times Int\left(\Delta \right)$ where Int(·) represents the interior.*

In simulations, we fixed k=2 to reduce the computational complexity and fixed the regularizer ζ=0.01 for *X* of diameter 1 and scales proportional with $\mathrm{diam}{\left(X\right)}^{k}$ (see next section). Such a choice for ζ is not only small enough to reduce the error between the EOT estimation $W_{k,\zeta }$ and the true $W_{k}$, but also ensures that $e^{-g(x,y)/\zeta }$ and its byproduct in the SK are distinguishable from 0 in a *double* format.

**Examples of discretization:** We demonstrated our algorithm on the following:

E.g., (1). μ is the uniform distribution on X=[0,1].

E.g., (2) μ is the mixture of two truncated normal distributions on X=[0,1], and the PDF is f(x)=0.3ϕ(x;0.2,0.1)+0.7ϕ(x;0.7,0.2), where ϕ(x;ξ,σ) is the density of the truncated normal distribution on [0,1] with the expectation ξ and standard deviation σ.

E.g., (3) μ is the mixture of two truncated normal distributions on $X={\left[0,1\right]}^{2}$, where the two distributions are ϕ(x;0.2,0.1)ϕ(y;0.3,0.2) of weight 0.3 and ϕ(x;0.7,0.2)ϕ(y;0.6,0.15) of weight 0.7.

Let N=100 for all plots in this section. Figure 2a–c plots the discretizations ($\mu_{m}$) for E.g., (1)–(3) with m=5,5, and 7, respectively.

Figure 2f illustrates the convergence rate of $W_{k,\zeta }^{k}\left(\mu ,\mu_{m}\right)$ versus the SGD steps for Example (2) with $\mu_{m}$ obtained by a 5-point EDOT. Figure 2d,e plot the entropy-regularized Wasserstein $W_{k,\zeta }^{k}\left(\mu ,\mu_{m}\right)$ versus *m*, thereby comparing EDOT and naive sampling for Examples (1) and (2). Here, the $\mu_{m}$s are: (a) from the EDOT with 3≤m≤7 in Example 1 and 3≤m≤8 in Example 2, which are shown by ×s in the figures. (b) from naive sampling, which is simulated using a Monte Carlo of volume 20,000 on each size from 3 to 200. Figure 2d,e demonstrate the effectiveness of the EDOT: as indicated by the orange horizontal dashed line, even 5-point EDOT discretization in these two examples outperformed 95% of the naive samplings of size 40, as well as 75% of the naive samplings of size over 100 (the orange dash and dot lines).

**An example of a transference plan:** In Figure 3a, we illustrate the efficiency of the EDOT on an OT problem: X=Y=[0,1], where the marginal μ and ν are truncated normal (mixtures), and μ has two components (shown in red curve on the left), while ν has only one component (shown in red curve on the top). The cost function is the squared Euclidean distance, and λ=ζ=0.01.

The left of Figure 3a shows a 5×5 EDOT approximation with $W_{k,\zeta }^{k}\left(\mu ,\mu_{5}\right)=4.792\times 10^{-3}$, $W_{k,\zeta }^{k}\left(\nu ,\nu_{5}\right)$$=5.034\times 10^{-3}$, and $W_{k,\zeta }^{k}\left(\gamma ,\gamma_{5,5}\right)=8.446\times 10^{-3}$. The high density area of the EOT plan is correctly covered by EDOT estimating points with high weights. The right shows a 25×25 naive approximation with $W_{k,\zeta }^{k}\left(\mu ,\mu_{7}\right)=5.089\times 10^{-3}$, $W_{k,\zeta }^{k}\left(\nu ,\nu_{7}\right)=2.222\times 10^{-2}$, and $W_{k,\zeta }^{k}\left(\gamma ,\gamma_{7,7}\right)=2.563\times 10^{-2}$. The points of the naive estimating with the highest weights missed the region where the true EOT plan was of the most density.

## 5. Methods of Improvement

I. Adaptive EDOT: The computational cost of a simple EDOT increases with the dimensionality and diameter of the underlying space. Discretization with a large *m* is needed to capture higher dimensional distributions, which result an increase in parameters for calculating the gradient of $W_{k,\zeta }^{k}$: md for the $y_{i}$ positions and m−1 for the $w_{i}$ weights. Such an increment will not only increase the complexity in each step, but also require more steps for the SGD to converge. Furthermore, the calculation will have a higher complexity (O(mN) for each normalization in Sinkhorn).

We proposed to reduce the computational complexity using a “divide and conquer” approach. The Wasserstein distance took the *k*-th power of the distance function dX|k as a cost function. The locality of distance dX| made the solution to the OT/EOT problem local, meaning that the probability mass was more likely to be transported to a close destination than to a remote one. Thus, we can “divide and conquer”—thereby cutting the space *X* into small cells and solve the discretization problem independently.

To develop a “divide and conquer” algorithm, we need: (1) an adaptive dividing procedure that is able to partition $X=X_{1}⊔\cdots ⊔X_{I}$, which balances the accuracy and computational intensity among the cells; (2) to determine the discretization size $m_{i}$ and choose a proper regularizer $\zeta_{i}$ for each cell $X_{i}$. The pseudocodes for all variations are shown in the Appendix C Algorithms A2 and A3.

Choosing size *m*: An appropriate choice of $m_{i}$ will balance contributions to the Wasserstein among the subproblems as follows: Let $X_{i}$ be a manifold of dimension *d*, let $\mathrm{diam}(X_{i})$ be its diameter, and let $p_{i}=\mu \left(X_{i}\right)$ be the probability of $X_{i}$. The entropy-regularized Wasserstein distance can be estimated as $W_{k,\zeta }^{k}=O\left(p_{i}m_{i}^{-k/d}\mathrm{diam}{\left(X_{i}\right)}^{k}\right)$ [16,17]. The contribution to $W_{k,\zeta }^{k}\left(\mu ,\mu_{m}\right)$ per point in support of $\mu_{m}$ is $O(p_{i}m_{i}^{-(k+d)/d}\mathrm{diam}{\left(X_{i}\right)}^{k})$. Therefore, to balance each point’s contribution to the Wasserstein among the divided subproblems, we set $m_{i}\approx \frac{{\left(p_{i}\mathrm{diam}{\left(X_{i}\right)}^{k}\right)}^{d/(k+d)}}{\sum_{j=1}^{I}{\left(p_{j}\mathrm{diam}{\left(X_{j}\right)}^{k}\right)}^{d/(k+d)}}$.

Occupied volume (Variation 1):A cell could be too vast (e.g., large in size with few points in a corner), thus resulting in obtaining a larger $m_{i}$ than needed. To fix it, we may replace the $\mathrm{diam}(X_{i})$ above with $\mathrm{Vol}{\left(X_{i}\right)}^{1/d}$, where $\mathrm{Vol}(X_{i})$ is the occupied volume calculated by counting the number of nonempty cells in a certain resolution (levels in previous binary division). The algorithm (Variation 1) becomes a binary tree to resolve and obtain the occupied volume for each cell, then there is tree traversal to assign $m_{i}$.

Adjusting the regularizer ζ: In the $W_{k,\zeta }^{k}$, the SK on $e^{-g(x,y)/\zeta }$ is calculated. Therefore, ζ should scale with $d_{X}^{k}$ to ensure that the transference plan is not affected by the scaling of $d_{X}$. Precisely, we may choose $\zeta_{i}=\mathrm{diam}{\left(X_{i}\right)}^{k}\zeta_{0}$ for some constant $\zeta_{0}$.

The division: Theoretically, any refinement procedure that proceeds iteratively and eventually makes the diameter of each cell approach 0 can be applied for division. In our simulation, we used an adaptive kd-tree-style cell refinement in a Euclidean space $R^{d}$. Let *X* be embedded into $R^{d}$ within an axis-aligned rectangular region. We chose an axis $x_{l}$ in $R^{d}$ and evenly split the region along a hyperplane orthogonal to $x_{l}$ (e.g., cut square ${\left[0,1\right]}^{2}$ along the line x=0.5); thus, we constructed $X_{1}$ and $X_{2}$. With the sample set *S* given, we split it into two sample sizes $S_{1}$ and $S_{2}$ according to which subregion each sample was located in. Then, the corresponding $m_{i}$ and $\zeta_{i}$ could be calculated as discussed above. Thus, two cells and their corresponding subproblems were constructed. If some of the $m_{i}$ was still too large, the cell was cut along another axis to construct two other cells. The full list of cells and subproblems could be constructed recursively. In addition, another cutting method (variation 2) that chooses the most sparse point as a cutting point through a sliding window is sometimes useful in practice.

After having the set of subproblems, we could apply the EDOT for the solutions in each cell, then combine the solutions μmi(i)=∑j=1miwj(i)δyj(i)| into the final result μm:=∑i=1I∑j=1mipiwj(i)δyj(i)|.

Figure 3b shows the optimal discretization for the example in Figure 2c with m=30, which was obtained by applying the EDOT with adaptive cell refinement, or $\zeta =0.01\times {\mathrm{diam}}^{2}$.

II. On embedded CW complexes: Although the samples on space *X* are usually represented as a vector in $R^{d}$, inducing an embedding $X↪R^{d}$, the space *X* usually has its own structure as a CW complex (or simply a manifold) with a more intrinsic metric. Thus, if the CW complex structure is known, even piecewise, we may apply the refinement on *X* with respect to its own metric, whereas direct discretization as a subset in $R^{d}$ may result in a low expressing efficiency.

We now illustrate the adaptive EDOT by an example on a mixture normal distribution of a sphere mapped through stereographic projection. More examples of a truncated normal mixture over a Swiss roll and the discretization of a 2D optimal transference plan are detailed in the Section D.5.

On the sphere: The underlying space $X_{\mathrm{sphere}}$ is the unit sphere in $R^{3}$. $\mu_{\mathrm{sphere}}$ is the pushforward of a normal mixture distribution on $R^{2}$ by stereographic projection. The sample set $S_{\mathrm{sphere}}∼\mu_{\mathrm{sphere}}$ over $X_{\mathrm{sphere}}$ is shown on Figure 4 on the left. Consider a (3D) Euclidean metric on the $X_{\mathrm{sphere}}$ induced by the embedding. Figure 4a (right) plots the EDOT solution with refinement for $\mu_{m}$ with m=40. The resulting cell structure is shown as colored boxes.

To consider the intrinsic metric, a CW complex was constructed about a point on the equator as a 0-cell structure; the rest of the equator was constructed as a 1-cell, and the upper hemisphere and lower hemisphere were constructed as two dimension 2- (open) cells. We took the upper and lower hemispheres and mapped them onto a unit disk through stereographic projection with respect to the south and north pole, respectively. Then, we took the metric from spherical geometry and rewrote the distance function and its gradient using the natural coordinate on the unit disk. Figure 4b shows the refinement of the EDOT on the samples (in red) and the corresponding discretizations in colored points. More figures can be found in the Appendices.

## 6. Analysis of the Algorithms

In this section, we derive the complexity of the simple EDOT and the adaptive EDOT. In particular, we show the following:

**Proposition** **3.**
*Let μ be a (continuous) probability measure on a space X. A simple EDOT of size m has time complexity $O({\left(N+m\right)}^{2}mdL+NmL\log\left(1/\epsilon \right))$ and space complexity $O({\left(N+m\right)}^{2})$, where N is the minibatch size (to construct $\mu_{N}$ in each step to approximate μ), d is the dimension of X, L is the maximal number of iterations for SGD, and ϵ is the error bound in the Sinkhorn calculation for the entropy-regularized optimal transference plan between $\mu_{N}$ and $\mu_{m}$.*


Proposition 3 quantitatively shows that, when the adaptive EDOT is applied, the total complexities (in time and space) are reduced, because the magnitudes of both *N* and *m* are much smaller in each cell.

The procedure of dividing sample set *S* into subsets through the adaptive EDOT is similar to Quicksort; thus, the space and time complexities are similar. The similarity comes from the binary divide-and-conquer structure, as well as that each split action is based on comparing each sample with a target.

**Proposition** **4.**
*For the preprocessing (job list creation) for the adaptive EDOT, the time complexity is $O(N_{0}\log N_{0})$ in the best and average case and $O(N_{0}^{2})$ in the worst case, where $N_{0}$ is the total number of sample points, and the space complexity is $O(N_{0}d+m)$, or simply $O(N_{0}d)$ as $m\ll N_{0}$.*


**Remark** **2.**
*Complexity is the same as Quicksort. The set of $N_{0}$ sample points in the algorithm are treated as the “true” distribution in the adaptive EDOT, since, in the later EDOT steps for each cell, no further samples are taken, as it is hard for a sampler to produce a sample in a given cell. Postprocessing of the adaptive EDOT has O(m) complexity in both time and space.*


**Remark** **3.**
*For the two algorithm variations in Section 5, the occupied volume estimation works in the same way as the original preprocessing step, which has the same time complexity as before (by itself, since dividing must happen after knowing the occupied volume of all cells), but, with the tree built, the original preporcessing becomes a tree traversal and has (additional) time complexity $O(N_{0})$ and (additional) space complexity $O(N_{0})$ for the space storing occupied volume.*

*For details on choosing cut points with window sliding, the discussion can be seen in the Section C.5.*


**Comparison with naive sampling:** After having a size *m* discretization on *X* and a size *n* discretization on *Y*, the EOT solution (Sinkhorn algorithm) has time complexity O(mnlog(1/ϵ)). In the EDOT, two discretization problems must be solved before applying the Sinkhorn, while the naive sampling requires nothing but sampling.

According to Proposition 3, solving a single continuous EOT problem using a size *m* simple EDOT method may result in higher time complexity than naive sampling with an even larger sample size *N* (than *m*). However, unlike the EDOT, which only requires access to a distance function $d_{X}$ and $d_{Y}$ on *X* and *Y*, respectively, a known cost function c:X×Y→R is necessary for naive sampling. In real applications, the cost function may be from real world experiments (or from extra computations) done for each pair (x,y) in the discretization; thus, the size of discretized distribution is critical for cost control. $d_{X}$ and $d_{Y}$ usually come along with the spaces *X* and *Y*, respectively, and are easy to compute. An additional application of the EDOT is necessary when the marginal distributions $\mu_{X}$ and $\nu_{Y}$ are fixed for different cost functions; then, discretizations can be reused. Thus, the cost of discretization is calculated one time, and the improvement it brings accumulates in each repeat.

## 7. Related Work and Discussion

Our original problem was the optimal transport problem between general distributions as samplers (instead of integration oracles). We translated that into a discretization problem and an OT problem between discretizations.

I. Comparison with other discretization methods: There are several other methods that generate discrete distributions from arbitrary distributions in the literature, which are obtained via semi-continuous optimal transport where the calculation of a weighted Voronoi diagram is needed. Calculating the weighted Voronoi diagrams usually requires 1. that the cost function be a squared Euclidean distance and 2. the application of Delaunay triangulation, which is expensive in more than two dimensions. Furthermore, semi-continuous discretization may only optimize one aspect between the position and weights of the atoms, and this process is mainly based on [18] (the optimized position) and [19] (the optimized weights).

We mainly compared the prior work of [18], which focuses on the barycenter of a set of distributions under the Wasserstein metric. This work resulted in a discrete distribution called the Lagrangian discretization, which is of the form $\frac{1}{m}\sum_{i=1}^{m}\delta_{x_{i}}$ [2]. Other works, such as [20,21], find barycenters but do not create a discretization. Refs. [19,22] studied the discrete estimation of a 2-Wasserstein distance locating discrete points through a clustering algorithm *k-means++* and a weighted Voronoi diagram refinement, respectively. Then, they assigned weights and made them non-Lagrangian discretizations. Ref. [19] (comparison in Figure 5)  roughly followed a “divide-and-conquer” approach in selecting positions, but the discrete positions were not tuned according to Wasserstein distance directly. Ref. [22] converged as the number of discrete points increased. However, it lacked a criterion (such as the Wasserstein in the EDOT) to show that the choice is not just one among all possible converging algorithms, but, rather, it is a special one.

By projecting the gradient in the SGD to the tangent space of the submanifold $X^{m}\times \left\{e_{m}/m\right\}=\left\{\frac{1}{m}\sum \delta_{x_{i}}\right\}$, or by equivalently fixing the learning rate on the weights to zero, the EDOT can estimate a Lagrangian discretization (denoted by EDOT-Equal). A comparison among the methods is held on the map of the Canary islands, which is shown in Figure 6. This example shows that our method can get a similar result using Lagrangian discretization as the methods in the literature, while, in general, this type of EDOT can work better.

Moreover, the EDOT can be used to solve barycenter problems.

Note that, to apply adaptive EDOT for barycenter problems, compatible divisions of the target distributions are needed (i.e., a cell A from one target distribution transports onto a discrete subset *D* thoroughly, and *D* transports onto a cell B from another target distribution, etc.).

We also tested these algorithms on discretizing gray/colored scale pictures. The comparison of discretization with points varying from 10 to 4000 for a kitty image between EDOT, EDOT-equal, [18] and estimations of their Wasserstein distances to the original image are shown in Figure 7 and Figure 8.

Furthermore, the EDOT may be applied on RGB channels of an image independently, which then combine plots of discretizations in the corresponding color. The results are shown in Figure 1 at the beginning of this paper.

Lagrangian discretization may have a disadvantage in representing repetitive patterns with incompatible discretization points.

In Figure 9, we can see that discretizing 16 objects with 24 points caused weight incompatibility locally for the Lagrangian discretization, thus making points locate between objects and increasing the Wasserstein distance. With the EDOT, the weights of points that lie outside of the blue object were much smaller. The patterned structure was better represented by the EDOT. In practice, patterns often occur as part of the data (e.g., pictures of nature), and it is easy to get an incompatible number in Lagrangian discretization, since the equal weight-requirement is rigid; consequently, patterns cannot be properly captured.

II. General *k* and deneral distance $d_{X}$: Our algorithms (Simple EDOT, adaptive EDOT, and EDOT-Equal) work for a general choice of parameter k>1 and $C^{2}$ distance $d_{X}$ on *X*. For example, in Figure 4 part (b), the distance used on each disk was spherical (arc length along the big circle passing through two points), which could not be isometrically reparametrized into a plane with Euclidean metrics because of the difference in curvatures.

III. Other possible impacts: As the OT problem widely exists in many other areas, our algorithm can be applied accordingly, e.g., the location and size of supermarkets or electrical substations in an area, or even air conditioners in the rooms of supercomputers. Our divide-and-conquer methods are suitable for solving these real-world applications.

IV. OT for discrete distributions: Many algorithms have been developed to solve OT problems between two discrete distributions [3]. Linear programming algorithms were first developed, but their applications have been restricted by high computational complexity. Other methods such as [23], with a cost of form c(x,y)=h(x−y) for some *h*, which applies the “back-and-forth” method by hopping between two forms of a Kantorovich dual problem (on the two marginals, respectively) to get a gradient of the total cost over the dual functions, usually solve problems with certain conditions. In our work, we chose to apply an EOT developed by [8] for an estimated OT solution of the discrete problem.

## 8. Conclusions

We developed methods for efficiently approximating OT couplings with fixed size m×n approximations. We provided bounds on the relationship between a discrete approximation and the original continuous problem. We implemented two algorithms and demonstrated their efficacy as compared to naive sampling and analyzed computational complexity. Our approach provides a new approach to efficiently compute OT plans.

## Figures and Tables

**Figure 1 entropy-25-00839-f001:**
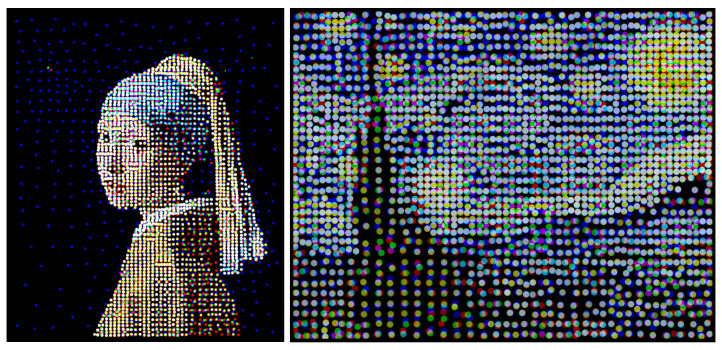
Discretization of “Girl with a Pearl Earring” and “Starry Night” using EDOT with 2000 discretization points for each RGB channel. k=2, $\zeta =0.01\times {\mathrm{diam}}^{2}$.

**Figure 2 entropy-25-00839-f002:**
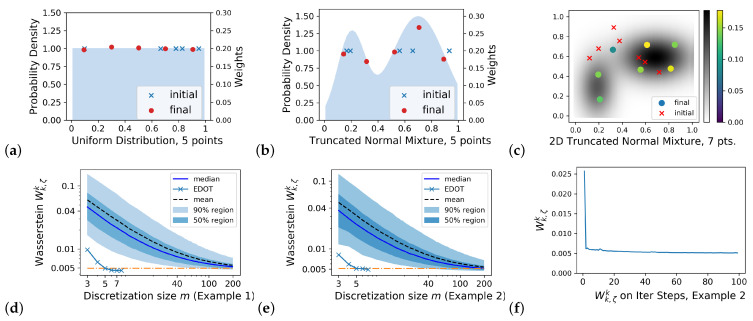
(**a**–**c**) Plots of EDOT discretizations of the Examples (1)–(3). In (**c**), the *x*-axis and *y*-axis are the 2D coordinates, and the probability density of μ and weights of $\mu_{m}$ are encoded by color. (**d**,**e**) show comparison between EDOT and i.i.d. sampling for Examples (1) and (2). EDOT are calculated with m=3 to 7 (3 to 8). The 4 boundary curves of the shaded region are 5%-, 25%-, 75%-, and 95%-percentile curves; the orange line represents the level of m=5; (**f**) plots the entropy regularized Wasserstein distance $W_{k,\zeta }^{k}\left(\mu ,\mu_{m}\right)$ versus the SGD steps for Example (2) with $\mu_{m}$ optimized by 5-point EDOT. ζ=0.01 in all cases.

**Figure 3 entropy-25-00839-f003:**
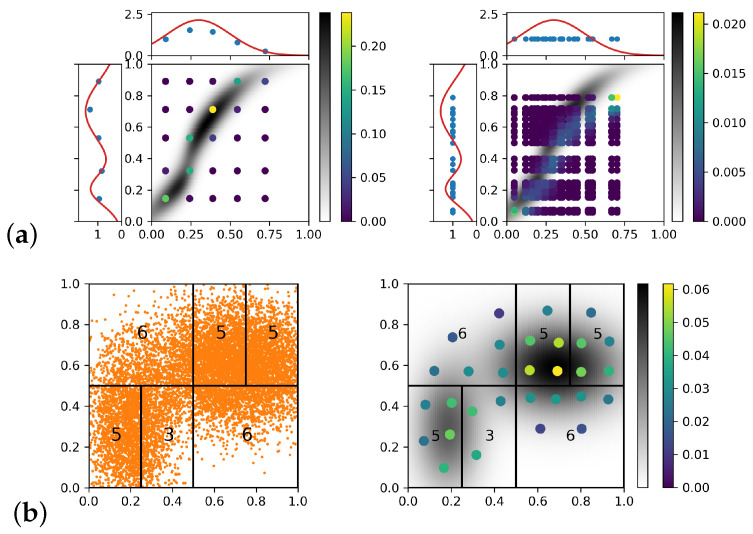
(**a**): Approximation of a transference plan. **Left**: 5×5 EDOT approximation. **Right**: 25×25 naive approximation. In both figures, magnitudes of each point is color coded, the background grayscale density represents the true EOT plan. (**b**): An example of adaptive refinement on a unit square. Left: division of 10,000 sample *S* approximating a mixture of two truncated Gaussian distributions and the refinement for 30 discretization points. Number of discretization points assigned to each region is marked by black numbers. E.g., upper left regaion needs 6 points. Right: the discretization optimized locally and combined as a probability measure with k=2.

**Figure 4 entropy-25-00839-f004:**
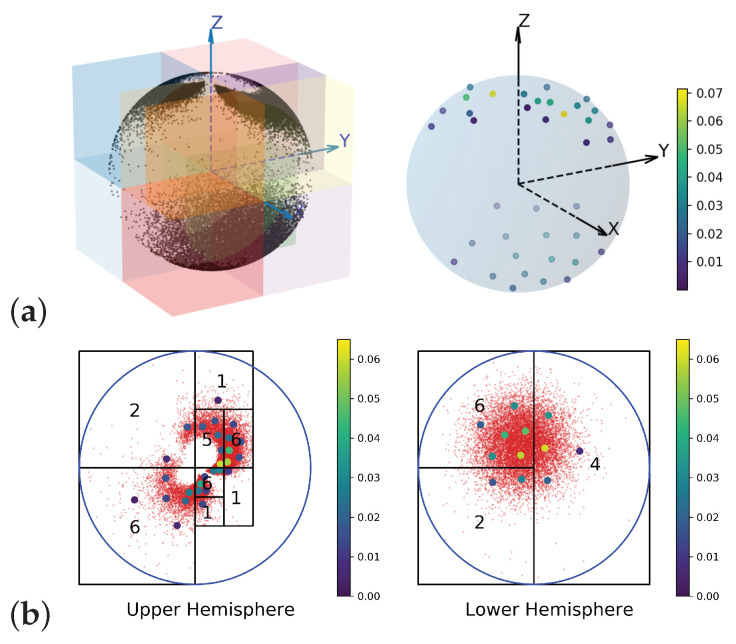
(**a**) **Left**: 30,000 samples from $\mu_{sphere}$ and the 3D cells under divide-and-conquer algorithm **Right**: 40-point EDOTs in 3D. (**b**) The 40-point CW-EDOTs in 2D. Red dots: samples, other dots: discrete atoms with weights represented in colors. **Left**: upper hemisphere. **Right**: lower hemisphere, stereographic projections about poles. $\zeta =0.01\times {\mathrm{diam}}^{2}$.

**Figure 5 entropy-25-00839-f005:**
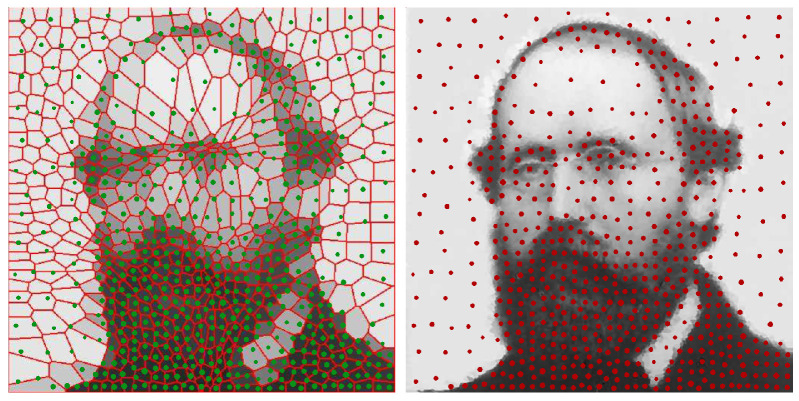
EDOT of an example from [19]. Potrait of Riemann, discretization of size 625. **Left**: green dots show position and weights of EDOT discretization (same as right); cells in background are discretization of the same size in the original [19]. **Right**: A size 10,000 discretization from [19]; we directly applied EDOT to this picture, treating it as the continuous distribution. $\zeta =0.01\times {\mathrm{diam}}^{2}$.

**Figure 6 entropy-25-00839-f006:**
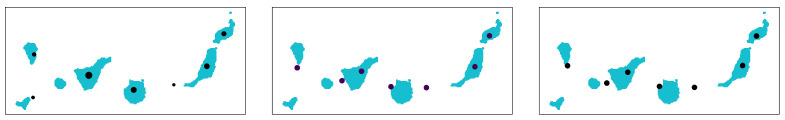
A comparison of EDOT (**left**), EDOT-Equal (**mid**), and [18] (**right**) on the Canary islands, treated as a binary distribution with a constant density on islands and 0 in the sea. Discretizations for each method is shown by black bullets. Wasserstein distances: EDOT: $W_{0.005}^{2}=0.02876$, EDOT-Equal: $W_{0.005}^{2}=0.05210$, Claici: $W_{0.005}^{2}=0.05288$. Map size is 3.13×1.43.

**Figure 7 entropy-25-00839-f007:**
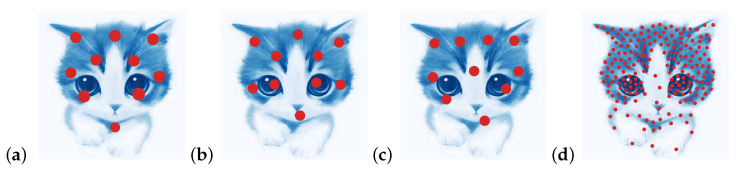
Discretization of a kitty. Discretization by each method is shown in red bullets on top of the Kitty image. (**a**) EDOT, 10 points, $W_{0.001}^{2}=0.009176$, radius represents weight; (**b**) EDOT-Equal, 10 points, $W_{0.001}^{2}=0.008960$; (**c**) [18], 10 points, $W_{0.001}^{2}=0.009832$; (**d**) [18], 200 points. Figure size 1×1, $\zeta =0.01\times {\mathrm{diam}}^{2}$.

**Figure 8 entropy-25-00839-f008:**
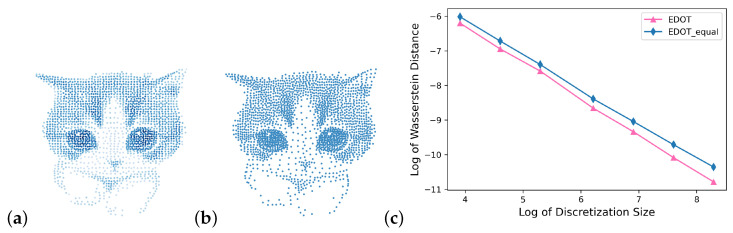
2000-Point Discretizations, (**a**). EDOT (weight plotted in color), (**b**). EDOT-Equal, (**c**). Relations between $\log(W^{2})$ and logm (all with divide and conquer); it can be seen that the advantage of $W_{\mathrm{EDOT}}$ over $W_{\mathrm{Equal}}$ grows with the size of discretization.

**Figure 9 entropy-25-00839-f009:**
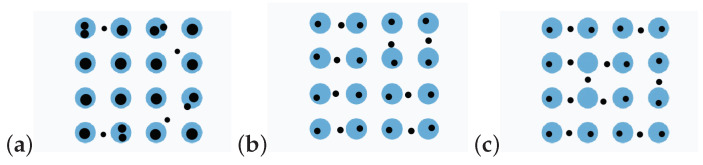
Discretization of 16 blue disks in a unit square with 24 points (black). (**a**) EDOT, $W_{\zeta }^{2}=0.001398$; (**b**) EDOT-Equal, $W_{\zeta }^{2}=0.002008$; (**c**) [18], $W_{\zeta }^{2}=0.002242$. $\zeta =10^{-4}$. Figure size is 1×1.

## Data Availability

Not applicable.

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
