# Peer review of "Efficient Discretization of Optimal Transport"

_entropy, 2023, doi:10.3390/e25060839_

Round 1

Reviewer 1 Report

The topic of optimal transport is super important and work of the authors is absolutely fabulous! I recommend accepting the paper as it is. 

Author Response

We thank the reviewer for the comments!

Some changes according to other reviewer's suggestions are made, after a more careful revision.

Reviewer 2 Report

There are two major issues I see with this paper:

1. The reader is promised a method suitable for large scale applications, but the evidence is presented in use-cases which are not.
- if large-scale means that the data lives in high-dimensional spaces, then 2D color images do not fit the bill
- if large-scale means "lots of data points", then 2D images may fit the bill. Nevertheless they are very structured and have sparse representation making undersampling / multiscale methods quite efficient for discretization. In this case "large scale but sparse because easily discretized" is an important nuance which needs to made clear.

2. Insufficient empirical evidence. I recognize two papers in the literature that the method needs to be compared to:

- Quentin Merigot,  A multiscale approach to optimal transport. Here, discretization/quantization is done thanks to Lloyds' algorithm for which there are parallel, online and large-scale implementations. Benchmarks are also done. And even with that level of care, I am not entirely convinced that it works well in practice -- especially if one aims for high machine precision. I have tried using such methods.

- Matt Jacobs and Flavien Leger. A fast approach to optimal transport: The back-and-forth method.  This is definitely the fastest implementation for images. Indeed, because of the grid structure and separable structure of the distance, many tricks are available. The method is brutal and unrefined with C code available, yet very efficient. Furthermore, it shows that in such contexts, there is no real need for the Sinkhorn epsilon regularization.

--------------------------------

More comments:

I. Eq. (2) does not hold "Without loss of generality". It is fine and satisfied for most L^p distances as you mention. But certainly not as general as an arbitrary cost. And why does the constant A disappear in A2? No power k? So I needed to recall the condition from somewhere else?

II. It is my understanding that you designed Proposition 2.1 to say that discretized EOT is controlled by the errors in the discretization. However, the constant C_1 is completely implicit. Without knowing how it behaves with \lambda and \zeta makes it look impractical.
If a sharp constant is hard to obtain, one could settle for empirical evidence that such a control is nice. And the rest of the paper focuses on the discretizations and does not control for the error in that original EOT problem.
But I suspect the control to be not nice unless \lambda and \zeta are large.

Here there is the same issue in Merigot's paper: good approximation of the measures may not yield good approximations of the transport plans, if one is not careful. For example, the well-known counter-example for smoothness of transport maps by Cafarelli shows that optimal couplings can be very rough although the initial measures are nice/can be nicely discretized.

I admit that is a hard problem. Yet one cannot simply ignore the issue.

III. In fact, as the rest of the paper is focused on discretizations, how does your algorithm compare to a classical quadrature approximation? Also, k-means / Lloyd would be a good sanity check. You mention k-means but I see no benchmark.

IV. It is well-known that small lambda and small zeta is when all hell breaks loose, yet this issue is swept under the carpet. How do you tune them?
As mentioned in point II, there are problems there.

Author Response

We thank the reviewer a lot for the precious comments and questions. For all that we can checkout, we edited our manuscript accordingly, the manuscript is ready to be uploaded while we are writing this response.

Majors:
1. We really appreciate the reviewer for pointing out this issue. We planned to mention that the discretization and OT algorithm can be applied not only to AI problems but also to many other real-world problems, where large scale problems, like locating supermarkets / electrical substations together with their sizes, exists when we consider a large range with complicated structures. However, we forgot to mentioned it when we wrote the paper, but alluded to it in the abstract. We now mentioned that in section 7, briefly, as the main goal of this paper is not those applications.

2. Empirical evidence: We added a comparison to [1. Merigot], by decorating on one of their figure, portrait of Riemann, since we were unable obtain their source code. We see different styles in the comparison, our version usually distributes more "even" or "uniform" on a near-uniform region (e.g. the coat on portrait of Riemann), while the [1. Merigot] generated a dense but not so even discretization. This is because [1. Merigot] can perform discretization, but do not have control on location (optimization is on weights), which is similar to our other cited paper [Beugnot, et. al. Sample complexity of Sinkhorn diverges] that uses another method to ``locate'' the positions and optimize weights. [Claici, et. al. Stochastic Wasserstein Barycenters] fixes weights to be uniform and move the positions, which is part of our algorithm's ability. We do have our own restriction, we trade generality and smoothness by exactness: we use estimated entropy-regularized Wasserstein distance instead the singular version.

The second paper [2. Jacobs] is good and very fast for images in performing OT algorithm, but restricts the cost function, $c(x,y)=h(x-y)$. After discretization, we can definitely choose to apply the method in [2. Jacobs] to perform discrete version of OT problem in suitable situations. Furthermore, sometimes discretization itself could become the goal, as mentioned in our response of last issue, our "byproduct" could be useful alone.

Comments:
I. Thank you for pointing out that the condition is not absolutely general, as $d_{X\times Y}$ is originally independent on $d_X$ and $d_Y$. But since we want to compare the Wasserstein distances on marginals (on $X$ and $Y$) and the Wasserstein distance on transport plan (on $X\times Y$), it is necessary to introduce some relation between $d_{X\times Y}$ and $(d_{X},d_{Y})$. We think one principle should be that if we require $d_{X\times Y}$ restricted on $X$-slices and $Y$-slices be the metrics over them, respectively. To illustrate, let's think about two atomic distributions $\delta_x$, and $\delta_{x'}$ on $X$, transported to one general distribution on $Y$. The above principle is the only way to guarantee the Wasserstein distance between the two transport plans (from the two atomic distributions) equals the Wasserstein distance between $\delta_x$ and $\delta_{x'}$, which is further $dX(x, x')$. Under the above principle, the condition we provide in the paper can be induced by triangle inequality, which is general. However, we still changed our language there in order to avoid misunderstandings in the updated manuscript.

II. It is true that we could not control the constant in theory. We thank the reviewer for point out a possible solution. The original version contains 5 random examples in Section 4 and Appendix D following the original experiments, in three of which the constant is 1 and in the other two the constant is smaller than 2. This problem is interesting, and it might be valuable to systematically study this problem as a new project.

III. The paper [Beugnot, et.al. Improving Approximate Optimal Transport Distances using Quantization] applies kmeans++ and calculated the Voronoi diagram to "discretize" the distribution, but there is no improvement step in their paper, so the locations are totally random via applying kmeans++ directly (which is a one-round sampling, but requires the calculation of distances a lot of times if not improved by additional structures like kd-tree). We did not make such a comparison because no optimization step was applied in that paper on discretization and it is even difficult to find a typical representative of the discrete distributions since the position changes every time.

IV. We agree that the introduction of lambda and zeta are the prices we paid in order to get smoothness and free from restrictions on discretization ability / cost / dimension, etc. These parameters, lambda and zeta, are kept as small as possible throughout our work, in order to minimize their consequences on the optimization quality. Our zeta was taken 0.001, 0.005, 0.01 and 0.04 in experiments, usually below 0.01. This is partially why we would rather applying divide and conquer than optimizing directly, as small tiles can accept smaller zeta value.

Reviewer 3 Report

The manuscript Efficient Discretization of Optimal Transport deals with discretization and its optimization problem. Overall, the work is interesting, and the author’s assumptions are well-funded and justified. However, some points require revision and improvement, namely:

1.       The authors create too many breaks in the text, namely paragraphs. It identified 9 wrongly placed paragraphs. Please revise the entire manuscript.

2.       L1 – “X × Y is discretized through discretizations of X and Y to respect marginal structure in OT.” Please revise.

3.       Fig2 (6) the Y axis should begin at zero.

4.       L144-149 this explanation is not clear, please revise.

5.       L188 Please provide evidence that the “divide and conquer” approach will not converge to a local minimum.

6.       L209-221 The division, it is not clear how the authors ensure that the best solution is not overpassed in the division process.

7.       The figure’s font should be increased to improve readability.

8.       The goal of the work is to present an optimization algorithm; however, no results are presented that compare the time gains or computational cost.

The work is interesting with some unique ideas and relevant novelty. The missing demonstration of non-local minimum and the computational gains are small setbacks that can be easily overcome before publication.

Author Response

We thank the reviewer a lot for the precious comments and questions. For all that we can checkout, we edited our manuscript accordingly, the manuscript is ready to be uploaded while we are writing this response.

  1. We revised accordingly. Thanks for pointing out.
  2. By that we meant that the points on $X\times Y$ is in Grid-like structure instead of an arbitrary point set. We've checked it out accordingly.
  3. We updated the figure.
  4. We changed accordingly.
  5. Sorry, we cannot prevent the algorithm from converging to local minima, with or without divide and conquer method. This is actually why we name our work ``Efficient Discretization of OT'' instead of ``Optimal Discretization of OT''.
  6. Same as 5. there is no guarantee of the optimiality of divide and conquer. But we have tried our best in pre-estimating where to divide, which is the main challenge over that part.
  7. We enlarged figures accordingly.
  8. Generally speaking, we think optimal discretization is still too hard for not only us but also all other papers we cited. So we turned to the next best thing, which is to make a trade-off between a good but not perfect answer and the cost in computation, meaning the space and time complexities.

For the optimality problem, during our whole journey of this project we were looking for an optimal solution, until we realize that the problem (discretization problem, not the optimal transport) is not convex, similar to the difficulties people face in deep learning.

First, for two discrete distributions of n-atoms, the linear combination of them may not have n-atoms, but n^2-atoms in general (the linear interpolation of two n-atom distributions), so with fixed number of atoms, the set of distributions with movable atom positions even fails to be a convex set (under the geometry on the space of probability distributions on X)!

Second, we can construct an example with two local minima in the following way: consider a distribution μ of 4 atoms locating at corners of a near-square rectangle (ABCD), each of equal weight (0.25). Let's discretize it using a 2-atom distribution under Euclidean distance square ($d(x,y) = |x-y|^2$). Two local minima can be: 1. two atoms of weight 0.5 locating at midpoints of edges AB and CD, 2. two atoms of weight 0.5 locating at midpoints of edges AD and BC. A perturbation on the positions and weights of discretizations can show the local minimality. As ABCD is slightly not square, the two local minima correspond different Wasserstein distances.

Round 2

Reviewer 3 Report

In this new revision, the work has been improved, and became clear to follow the author's thoughts. I recommend the work for publication

Author Response

We appreciate the reviewer for patience and kindness in all the comments. We will do a final revision on language, and update the manuscript.